# Local deprivation predicts right-wing hate crime in England

**Margherita Belgioioso[1]* , Christoph Dworschak[2] , Kristian Skrede Gleditsch[3,4]**

**1** School of Politics and International Studies, University of Leeds, Leeds, United Kingdom, **2** Department of Politics, University of York, York, United Kingdom, **3** Department of Government, University of Essex, Colchester, United Kingdom, **4** Peace Research Institute Oslo, Oslo, Norway

☯ These authors contributed equally to this work.
* M.Belgioioso@leeds.ac.uk

**Data Availability Statement:** Data available at the Harvard Dataverse at https://doi.org/10.7910/DVN/GSRPQY.

**Funding:** Gleditsch is grateful for funding under the project "The Crime-Reducing Effect of Education", grant number 302445, from the Research Council

## Abstract

We argue that community deprivation can increase the risk of right-wing radicalization and violent attacks and that measures of local deprivation can help improve forecasting local hate crime rates. A large body of research stresses how experiences of deprivation can erode the perceived legitimacy of political leaders and institutions, increase alienation, and encourage right-wing radicalization and hate crime. Existing analyses have found limited support for a close relationship between deprivation and radicalization among individuals. We provide an alternative approach using highly disaggregated data for England and show that information on local deprivation can improve predictions of the location of right-wing hate crime attacks. Beyond the ability to predict where right-wing hate crime is likely, our results suggest that efforts to decrease deprivation can have important consequences for political violence, and that targeting structural facilitators to prevent far-right violence ex ante can be an alternative or complement to ex post measures.

## Introduction

Far-right violence is a major security concern in many high-income democracies, and the number of right-wing terrorist attacks has by one estimate increased by 250% over the period 2015–2019 [1]. Many have argued that social and economic deprivation can motivate political violence such as civil war and terrorist attacks [2], and the link between deprivation, radicalization, and violence seems particularly relevant to many nativist right-wing ideologies. Existing research has suggested limited evidence for a link between deprivation and radicalization or violence at the individual level [3, 4]. We argue that existing studies have often ignored the role of local community characteristics in fostering radicalization and increased risk of violent attacks, and that it may be possible to leverage information on local deprivation to better predict the risk of right-wing hate crime attacks.

In this article, we examine whether higher neighborhood-level deprivation is associated with a greater risk of far-right hate crime. Observing local deprivation can make individuals feel that their communities have been left behind, perceive political leaders and existing institutions as unresponsive, unhelpful, or illegitimate, and foster greater susceptibility to far-right ideas and frames of responsibility [5]. We provide a first study to systematically examine this

of Norway. The funders had no role in study design, data collection and analysis, decision to publish, or preparation of the manuscript.

**Competing interests:** The authors have declared that no competing interests exist.

relationship at the level of neighborhoods, using disaggregated data on deprivation across neighborhoods in England as well as the location of right-wing hate crime attacks. We train our models on data for 2015–2018 and find substantially more far-right attacks in more deprived local communities. We then show that information on local deprivation substantially improves forecasts of right-wing attacks over the period 2019–21 over a baseline model limited to other demographic and socio-economic information. This study demonstrates how it is possible to develop better models to forecast right-wing hate crime attacks by using local measures of deprivation. Our arguments on the mechanisms underlying our findings also suggest that policies aimed at decreasing deprivation could have important benefits for reducing political violence, and that efforts to target structural local drivers of radicalization beyond ideological beliefs could improve preventive counter-terrorist measures and help complement *ex post* interventions targeting people who are already radicalized or have already carried out attacks.

## Local deprivation and extremist political violence

Many theories of grievances and political violence argue that political and economic inequalities following group lines can generate resentment and political mobilization that increase the likelihood of resort to violence [2]. We believe that a similar process also applies to right-wing radicalization and violence at the local level. We focus on local socio-economic deprivation, by which we mean how average outcomes on relevant social and economic indicators for a local community compare to the average for other local communities. A local community is more deprived when it is characterized by low or worse outcomes compared to other local communities. High local deprivation can erode the perceived legitimacy of political leaders and central institutions, undermine trust, increase political and social alienation, and even lead to support for the use of violence [6, 7]. We posit that the local neighborhood is the main reference point for individual comparisons, shaping individual perceptions and beliefs about outcomes and distributions. The local neighborhood often forms the basis for community identification and the evaluations people make about whether outcomes or distributions are "fair" or not [8, 9]. Individuals who observe high deprivation in their proximate local environments are more likely to consider their communities as neglected by the state and other political authorities [10]. This form of political dissatisfaction is likely to be accompanied by declining confidence in established political channels and the ability to change outcomes or overcome injustices through conventional political participation.

Once individuals are politically disaffected after experiencing local deprivation, they can also become more susceptible to extremist right-wing ideologies. These often portray deprivation as a result of social and economic injustices where the central political establishments either actively conspire to undermine the interests of native communities or alternatively fail to adequately respond to their needs and interests [11]. Far-right narratives tend to present as a common theme "a collective sense of persecution [of the "native" population], presenting themselves as victims of societal oppression" [12, 13]. These narratives offer a sense of belonging to a group—encouraging the self-identification to a broader group of victimized native citizens. They provide a clear identification of responsibility and culpable agents, and offer narratives that dehumanize "outsiders" in ways that generate more polarized and hardened group identities and fuel fears of future victimization [14]. This can, in turn, increase the perceived legitimacy of political violence.

There is empirical evidence that perpetrators of hate crimes tend to carry out violence within their local neighborhood [15]. Perpetrators often seek out victims within their immediate vicinity, as 1) it is easier to identify targets they hold prejudice against in the local community [16], 2) access, planning, and execution might be easier for local targets [17], 3) local

targets often have symbolic significance for both perpetrators and targeted communities [18], and 4) local hate crimes often receive more attention in the immediate community, which in turn can be helpful for generating fear and exerting control over targets and recruiting activists to the cause [19].

Our concept of local deprivation focuses on the characteristic outcomes for local communities rather than individual outcomes or distributions within the community population. Note that if local community deprivation is the most salient focus for relevant influence for radicalization and attacks, then community deprivation does not necessarily translate to a systematic relationship between indicators of deprivation and radicalization at the individual level. The individuals who carry out attacks need not necessarily be relatively more deprived, even if their local community is deprived. A similar claim in the context of terrorism notes that one would not expect a clear relationship between deprivation and attacks if groups can select the most able among the pool of their potential supporters [20]. Therefore, neither highly geographically aggregated approaches nor individual level studies can conclusively evaluate the potential relationship between local deprivation and radicalization or the risk of hate crime.

Clearly not all individuals holding extremist attitudes will go on to endorse or commit violent attacks. The UK Prevent scheme, for example, has suggested a pyramid model of individual radicalization, outlined in Fig 1 [21, 22]. At the top of the pyramid is the subset of the fully radicalized individuals who actually carry out acts of terrorism political violence. These are a relatively small share of a larger set of individuals who hold some degree of sympathy for, or are receptive to, radical ideological beliefs. This larger, active group of "sympathizers" can

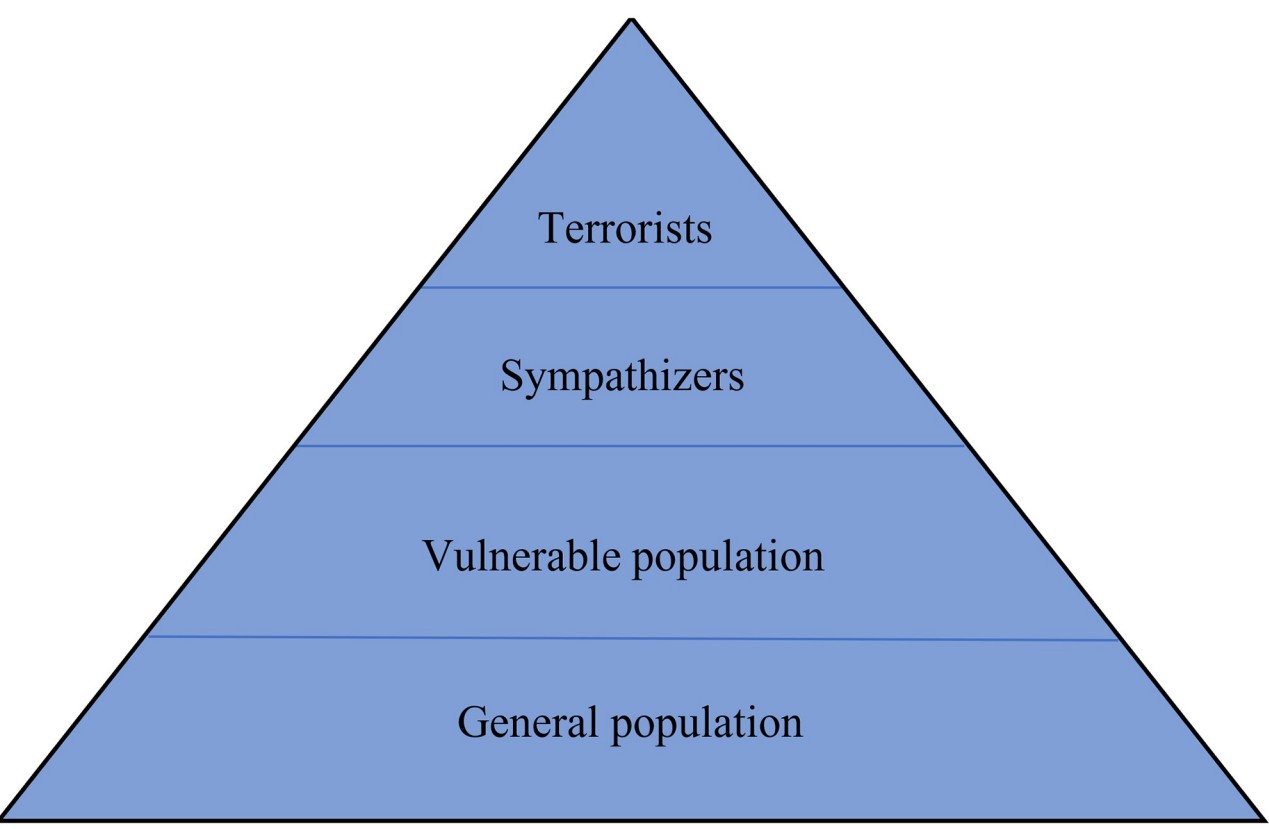

**Fig 1. Pyramid of radicalization.**

provide a pool of potential recruits or facilitate attacks, although it may be difficult to identify which sympathizers go on to carry out violent attacks. Below the active sympathizers are a larger group of uncommitted individuals who may be potentially susceptible to extreme ideological messages, i.e., the set considered as "vulnerable populations" in the UK prevent scheme. Our interest here is in seeing whether the notion of such a subset can be used to identify local contexts where there is a higher probability of radicalization and higher risk of right-wing attacks.

The UK Prevent Scheme explicitly "aims to stop people being drawn into terrorist-related activity" [21]. Existing work often takes a negative view on the prospects for forecasting right-wing hate crime on the grounds that it is very difficult to predict precisely which individuals among the potentially susceptible will actually become fully radicalized or carry out violent attacks [23]. But even if one cannot reliably predict which individuals will go on to engage in right-wing hate crime, we may still be able to identify what socio-economic characteristics make local contexts more prone to far-right attacks, if neighborhood-level deprivation plays an important role in inducing vulnerability to right-wing radicalization and a higher risk of local attacks.

Following this discussion, we propose that *higher neighborhood-level social and economic deprivation increases the risk of right-wing violence*, and that *measures of local deprivation can help improve predictions of right-wing hate crime.*

## Empirical data

We examine local deprivation and right-wing hate crime across England using data for Lower Layer Super Output Areas (LSOAs), the highest geographical resolution used in the UK census data. Each LSOA comprises between 400–1,200 households, with a resident population between 1,000 and 3,000 persons [23].

To capture hate crime, we obtained disaggregated *Hate Crime Statistics*, recorded by the UK Police for 2015–2021, and provided by the Crime Analysis Unit of the Home Office following a Freedom of Information request by the authors (FOI 69240). Hate crimes differ from other criminal offences in that they "express a number of socio-political objectives by targeting individuals based on their perceived group membership", including race, religion, disability, or sexual orientation and gender identity [24]. Hate crimes often resemble the more general concept of terrorist attacks, where the identity of the direct victims may be subordinate to the intention of indirectly imposing "terror" or cost on a different specific target such as the government or the general population. Hate crimes are often intended to convey a message to a broader audience beyond the direct victim, and often aim to instill fear, force a change in government policy, or help attract followers and recruits to the cause. Hate crimes also reflect key right-wing beliefs such as national chauvinism, xenophobia, and racism, where certain individuals are acceptable targets by virtue of their identity [25]. The perpetrators often seek out suitable victims within their immediate vicinity, as it is easier to identify targets and plan attacks within more familiar local environments, to gain access to these targets, and local targets often have a clear symbolic significance for perpetrators and audiences. Existing surveys find that hate crimes tend to be perpetrated locally, reflected in clear hate-crime hotspots [26]. In light of the above, we posit that the location of hate crimes is likely to reflect local right-wing radicalization and can be used to consider possible factors contributing to radicalization and the risk of hate crime attacks.

To capture local deprivation–which we conceptualize as and the extent of deviation from equality in the national distributions of goods, services, or negative outcomes (i.e., burdens),–we rely on indicators from the 2015 *English Indices of Deprivation* (IoD). These scores measure

relative local deprivation through a series of indices for 32,844 neighborhoods in England. We use the indices for deprivation in income, education, and environment as complimentary measures of local deprivation. We refer to the S1 Appendix for more detailed information on the construction of these indices and underlying data as well as the correlations between the different indices. As we only have access to the reported aggregate indices, we are unable to directly explore the inputs to the distinct indicators into a single local deprivation index in more detail, or to combine the input indicators in one general deprivation index. Fig 2 displays the distribution of hate crime and deprivation across LSOAs in England. The data underlying these deprivation indicators were gathered between 2012–2013. We believe the lag between our local deprivation measures and the post 2015 hate crime outcomes ensures that the measures are clearly prior to the response. Deprivation tends to be sticky, and comparisons to data measured at least two years before should be a reasonable window to consider the possible impact of past exposure to deprivation on outcomes. However, we acknowledge that the specific time lag used here is partly driven by available data. We do not have adequate coverage of hate crime data prior to 2015 or more recent deprivation indices. As such, we are restricted to a cross-sectional setup and are unable to consider empirically the temporal dimension of the exposure-violence relationship. Doing so would be a valuable contribution for future research to pursue.

Local deprivation and right-wing extremism may have common causes, and we condition on several other covariates to reduce confounding and better identify plausible partial relationships. We include data on total LSOA population size (logged), average age, percent of

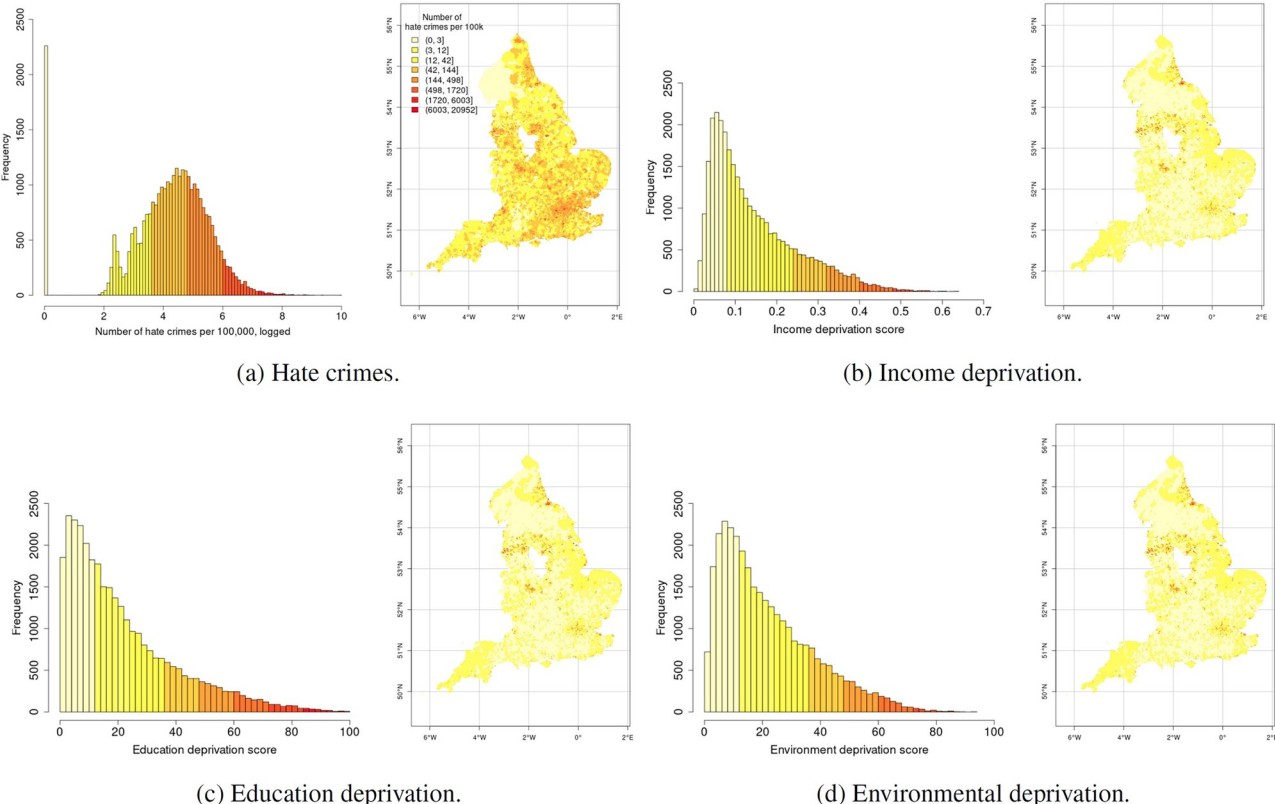

(a) Hate crimes.

(b) Income deprivation.

(c) Education deprivation.

(d) Environmental deprivation.

**Fig 2. Maps and histograms of the distributions of hate crime and deprivation indices.** Source for maps: Office for National Statistics licensed under the Open Government Licence v.3.0.

rural/urban population, level of unemployment, the Black and Minority Ethnic (BAME) population share, the share of people reporting to have Christian religion, and the share of people with Arab ethnicity. We also include spatial lags of our deprivation measures, capturing the mean values of all bordering neighborhoods. A key concern here is that many forms of political violence often diffuse, and attacks in one area may inspire similar efforts in nearby communities even in the absence of similar characteristics [27, 28]. However, if neighboring areas are likely to be similar to one another, then we overstate the effects of local deprivation without accounting for spatial clustering [28]. Finally, the reliability of hate crime reporting likely differs between police force districts, which also coincide with administrative districts. Therefore, we also include police force district fixed effects to partial out differences between districts. In the S1 Appendix we further explore the role of missingness and report additional analyses in which we omit neighborhoods without any hate crime reports from our sample (S2 Table in S1 Appendix) and model the zeros as a separate data generating process (S3 Table in S1 Appendix). We also report our main results with cluster-robust standard errors at the police district level (S5 Table in S1 Appendix). Our inferences remain the substantively unchanged. A list of all police force districts is included in S1 Table in S1 Appendix.

In Table 1 (below) we report the result for linear regression models of the log number of hate crimes per 100,000 between 2015–2018 against a baseline model with covariates only (1), models with deprivation measures only (2–4), and models combining deprivation measures and covariates (5–7). We find positive conditional associations for all the deprivation indicators, reflecting that we see more hate crime events in areas with more deprivation. These effects are visualized in Fig 3, based on bootstrapped observed values. Moreover, the full models 5–7 exhibit higher R2s and lower RMSEs and AICs than the baseline model 1, illustrating that information on deprivation contributes notably to accounting for variance in hate crimes.

**Table 1. Linear regression of logged hate crime 2015–2018 on deprivation.**

| | Model 1 | Model 2 | Model 3 | Model 4 | Model 5 | Model 6 | Model 7 |
|---|---|---|---|---|---|---|---|
| Income deprivation | | 7.226 | | | | 4.048 | | |
| | | (0.095) | | | | (0.181) | | |
| Living envir. deprivation | | | 0.029 | | | | 0.020 | |
| | | | (0.001) | | | | (0.001) | |
| Education deprivation | | | | 0.028 | | | | 0.008 |
| | | | | (0.001) | | | | (0.001) |
| Income depr. (neighb.) | | | | | 1.704 | | | |
| | | | | | (0.171) | | | |
| Living envir. depr. (neighb.) | | | | | | | −0.002 | |
| | | | | | | | (0.001) | |
| Education depr. (neighb.) | | | | | | | | 0.008 |
| | | | | | | | | (0.001) |
| Covariates | x | | | | x | x | x |
| Police district FE | x | | | | x | x | x |
| Num.Obs. | 32201 | 32201 | 32201 | 32201 | 32201 | 32201 | 32201 |
| R2 | 0.424 | 0.151 | 0.057 | 0.075 | 0.439 | 0.440 | 0.430 |
| RMSE | 1.46 | 1.77 | 1.86 | 1.84 | 1.44 | 1.44 | 1.45 |
| AIC | 115661 | 128048 | 131444 | 130822 | 115332 | 114758 | 114802 |

(Standard errors in parentheses).

(a)

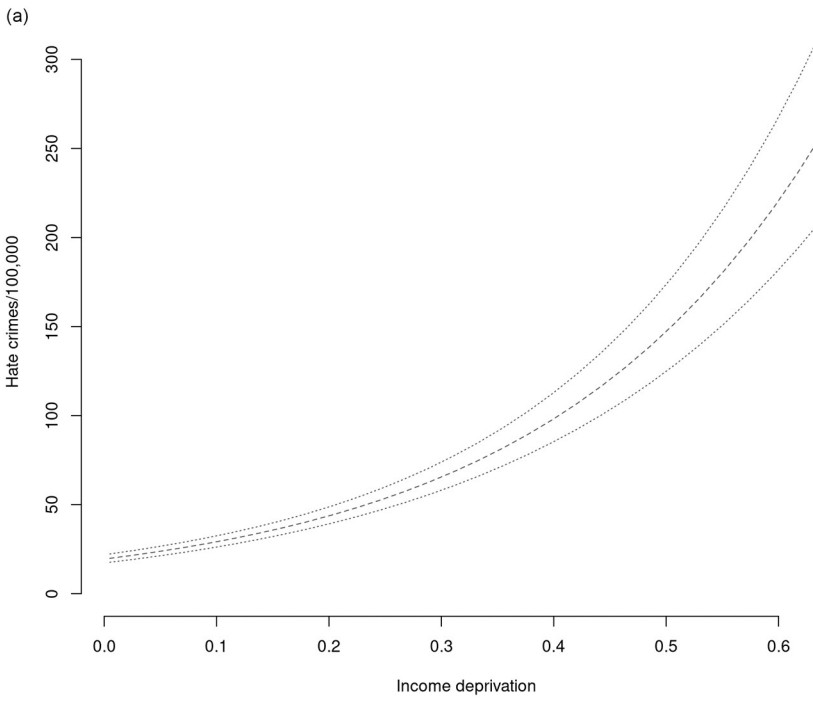

(b)

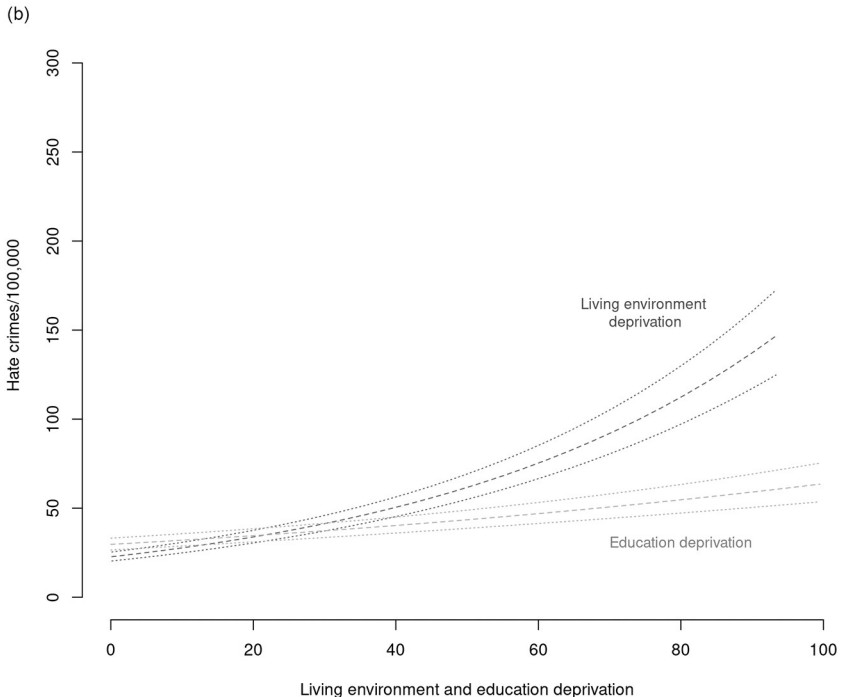

**Fig 3. Marginal effect visualizations of full models (5–7).**

The reported results from these regression models are all in-sample, and intended to demonstrate the conditional associations between the local deprivation indicators and hate crime in a manner that is simple to evaluate. However, in-sample estimation is prone to problems of overfitting. To see whether the apparent improvement in difference in performance also extends to improved forecasts further into the future, we conduct a prediction exercise using data for 2019–2021 as a hold-out period. Moreover, since a linear regression framework imposes a number of potentially strong assumptions about functional form, we use more flexible random forests, which also allow us to better consider the plausible importance of the different deprivation indicators simultaneously in the same model.

## Prediction

Turning from identification to prediction, we employ random forest models to benchmark the importance of local deprivation features for predicting hate crimes in England based on out-of-bag (OOB) node purity, comparing both the null model and full model feature space. Fig 4 suggests that structural information on the local BAME share and distribution of faith contribute the most to the prediction of hate crimes. In line with the model statistics of the in-sample analyses above, income deprivation visibly outperforms both education deprivation and living environment deprivation. The performance of the living environment and education deprivation indices for predicting hate crimes is comparable to that of other relevant covariates. One factor contributing to the performance of the income deprivation index may be that it takes

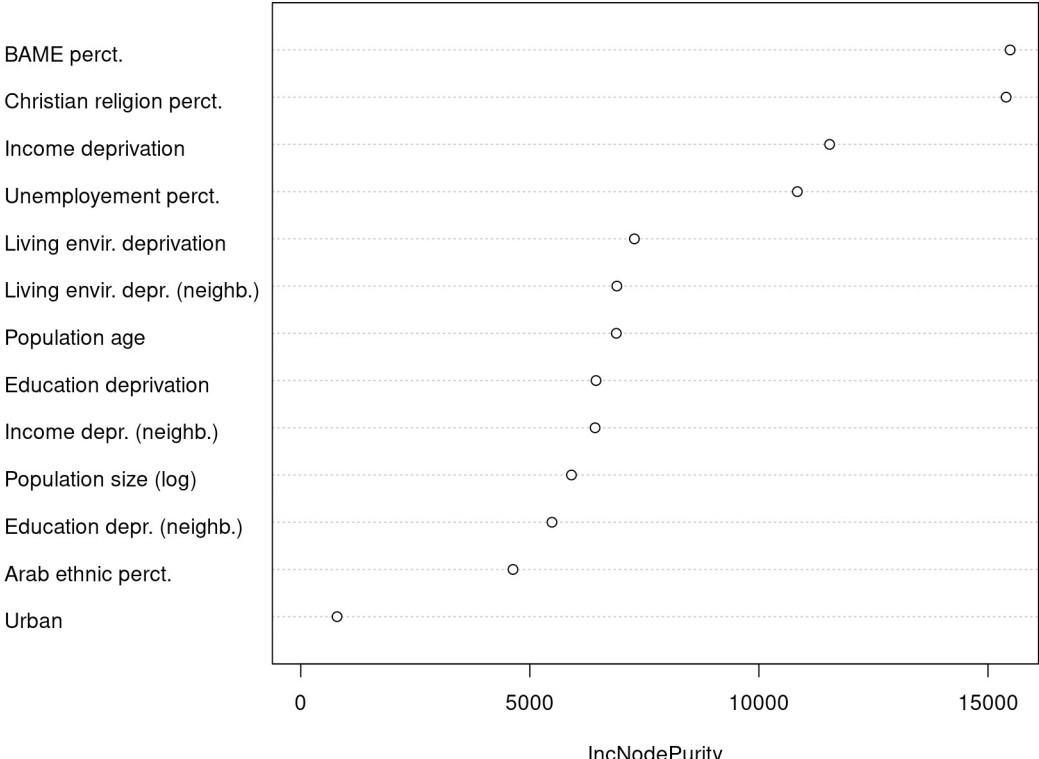

**Variable importance plot**

**Fig 4. Feature importance.**

the local number of asylum seekers into account as one of its components. In the S1 Appendix we further explore the role of ethnic diversity by interacting BAME share with our deprivation indicators, finding that the effect of income deprivation is indeed slightly moderated by ethnic diversity (S4 Table in S1 Appendix).

Turning to the substantive predictive performance of the measures, we then compare observed hate crime rates to those predicted by our models. Stratifying hate crime rate into eight bins, Fig 5 suggests that our model performs reasonably well in predicting the spatial distribution of hate crimes. In addition, comparing performance metrics of the full model to those of the null model that does not include information on deprivation, we find that the RMSE increases from 1.64 to 1.67, and the R2 decreases from 0.128 to 0.088. In other words, using demographic and structural information predicts about 9% of variance in the number of future hate crimes, while including information on deprivation increases this predicted variance to about 13%. Fig 5 suggests that deprivation provides important information to anticipate the predicted outcomes, in line with our argument that local deprivation breeds radicalization. For example, drawing on the deprivation measures attenuates the predicted number of hate crimes for many locations in the Cumberland unitary authority district in North-West England, but also increases numbers for several LSOAs in Durham in the North East, thereby moving the predictions closer to the observed rate of hate crimes.

Fig 6 shows how the full and null model compare to the observed data and to each other. The dotted line shows the line of perfect fit and the continuous line shows the regression lines. The null model clusters marginally more at low values, while the full model shows more spread across the range of observed hate crime. As may be expected, the plot also suggests that the model's prediction error follows a regression-to-the-mean pattern, in which values at the lower end of the scale tend to be overestimated and very high hate crime rates are underestimated. The high degree of dispersion, with most neighborhoods having low rates of hate crime and some having very high rates, leads both models to underestimate the rate of hate crime for more extreme cases.

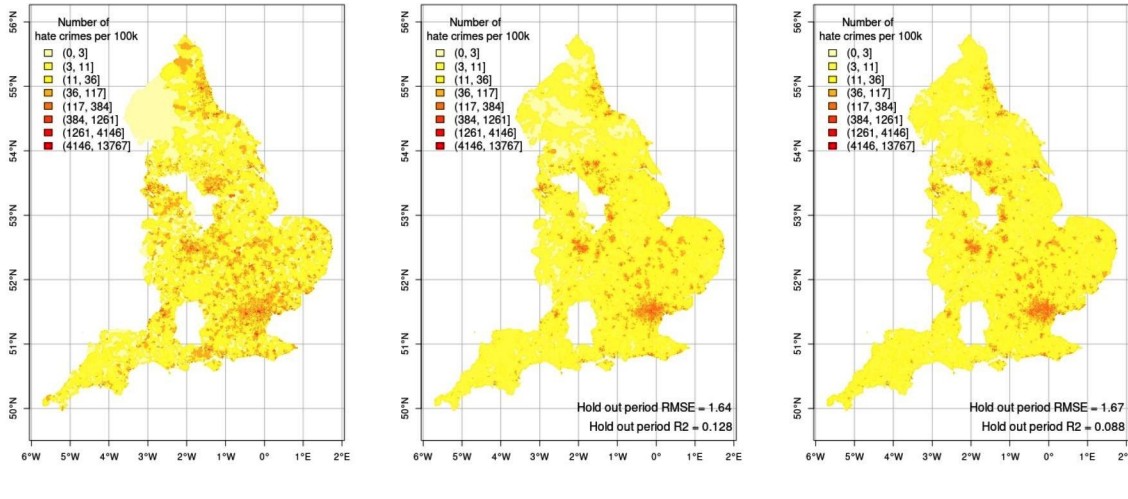

(a) Observed hate crimes, 2019-2021.    (b) Predicted hate crimes, full model.    (c) Predicted hate crimes, null model.

**Fig 5. Observed and predicted hate crimes rates, 2019–2021.** Source for maps: Office for National Statistics licensed under the Open Government Licence v.3.0.

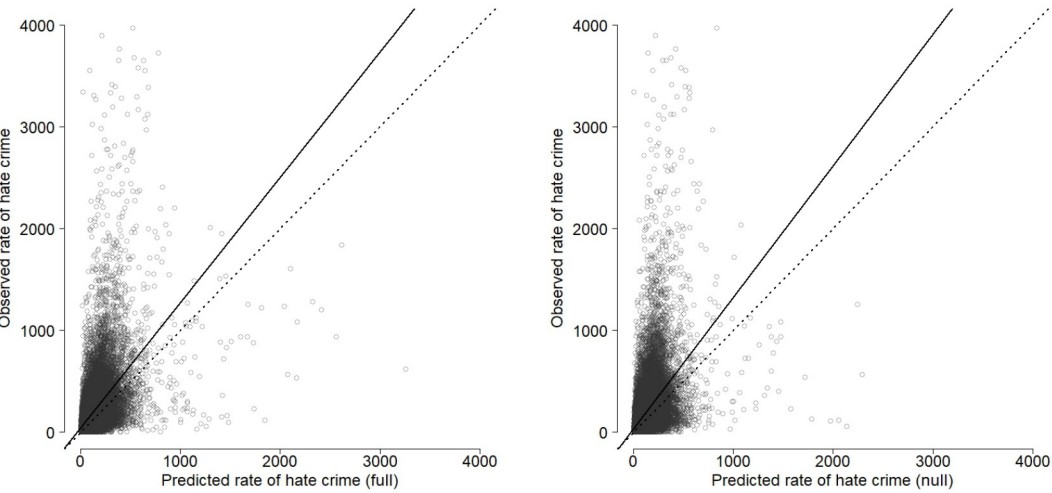

(a) Observed and predicted hate crimes, full model     (b) Observed and predicted hate crimes, null model.

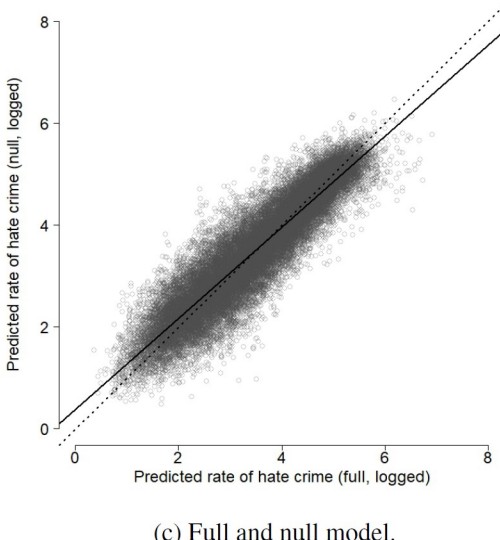

(c) Full and null model.

**Fig 6. Comparing predictions with and without deprivation measures.**

## Discussion and conclusion

We have examined the relationship between local deprivation and right-wing hate crime. Unlike previous studies that examine this relationship either at the individual level or at a highly aggregated level, our theory and analysis focus on local communities. We argue that neighborhood-level analyses allow more inferential power on the underlined links between deprivation and radicalization. Our results are consistent with the expectation that local deprivation is an important driver of right-wing radicalization and risk of far-right attacks. We also show that measures of local deprivation help improve forecasts of where right-wing hate crime can be expected.

Beyond their importance for theories of deprivation and radicalization and for forecasting where hate crimes are likely to occur, our results also suggest a potential scope for policies to

prevent future radicalization through reducing deprivation. Efforts to identify potential perpetrators of violent attacks to date appear to have low predictive ability and more general efforts to counter ideologies that may encourage attacks have been criticized for problematic profiling and demonization of specific communities [29]. The most common responses to countering hate crime focus on ex-post efforts to deradicalize, and target people who are already radicalized and either have carried out terrorist attacks or have attempted to do so. Our findings suggest that efforts to reduce local deprivation, beyond leading to other socially desired positive outcomes, may have important additional consequences for reducing hate crime. For example, policies focused on "levelling up", or improving the most deprived communities [30], could in addition to improving local living standards and decreasing inequalities, also help decrease the risk of far-right violence. However, we must stress that the ability to change deprivation through measures and intervention is a separate issue altogether, not directly examined in our analyses here. Direct plans for interventions must consider potential barriers for implementation and the possible unintended consequences as well as strategies to evaluate effectiveness. However, even though inequalities tend to be sticky and may be difficult to reduce, the efforts to reduce specific forms of inequality such as child poverty and deprivation among pensioners under the New Labour government over the period 1997–2010 are generally considered to have been remarkably effective, and provide a proof of concept for interventions to reduce deprivation and inequalities [31]. Above all, our work suggests that predicting right-wing violence in geographical hotspots based on local deprivation may be considerably easier than predicting the actions of individual perpetrators. It provides at least an intellectual rationale for exploring how neighborhoods can be a fruitful context to explore possible new preventive counter-terrorist measures.

## Supporting information

**S1 Appendix. Additional supporting information is provided in the document.**
(DOCX)

## Author Contributions

**Conceptualization:** Margherita Belgioioso, Christoph Dworschak, Kristian Skrede Gleditsch.

**Formal analysis:** Christoph Dworschak.

**Project administration:** Margherita Belgioioso.

**Software:** Christoph Dworschak.

**Supervision:** Kristian Skrede Gleditsch.

**Validation:** Christoph Dworschak.

**Visualization:** Christoph Dworschak.

**Writing – original draft:** Margherita Belgioioso.

**Writing – review & editing:** Christoph Dworschak, Kristian Skrede Gleditsch.

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
