## [Decision Letter · Decision Letter 0]

12 Jun 2023

PONE-D-23-08904Local deprivation predicts right-wing hate crime in EnglandPLOS ONE

Dear Dr. Gleditsch,

Thank you for submitting your manuscript to PLOS ONE. After careful consideration, we feel that it has merit but does not fully meet PLOS ONE’s publication criteria as it currently stands. Therefore, we invite you to submit a revised version of the manuscript that addresses the points raised during the review process.

Editor Comments:

The paper provides an intriguing and informative analysis. The referees have highlighted several concerns that require thorough attention and addressing from the authors. In addition, to further enhance its robustness, I would suggest that the authors incorporate lagged dependent variables in the regression model. This inclusion can account for the likelihood that previous instances of hate crimes contribute to subsequent hate crimes, potentially overshadowing the effect of local deprivation. Furthermore, it would be valuable for the authors to explore the possibility that the local deprivation effect can be understood within the context of micro-dynamic models that incorporate social networks as a mechanism for the diffusion of civil unrest behavior. For instance, Braha, D. (2012) examines this topic in their work "Global civil unrest: contagion, self-organization, and prediction" published in PloS One, 7(10), e48596. Discussing and referencing this work would provide additional insight into the dynamics of local deprivation and hate crime behavior.

We look forward to receiving your revised manuscript.

Best,

Dan

Prof. Dan Braha

Academic Editor

PLOS ONE

Journal Requirements:

"Gleditsch is grateful for funding under the project "The Crime-Reducing Effect of Education", funded under the FINNUT Programme of the Research Council of Norway."

4. We note that Figures 2 and 5  in your submission contain map/satellite images which may be copyrighted. All PLOS content is published under the Creative Commons Attribution License (CC BY 4.0), which means that the manuscript, images, and Supporting Information files will be freely available online, and any third party is permitted to access, download, copy, distribute, and use these materials in any way, even commercially, with proper attribution. For these reasons, we cannot publish previously copyrighted maps or satellite images created using proprietary data, such as Google software (Google Maps, Street View, and Earth). For more information, see our copyright guidelines: http://journals.plos.org/plosone/s/licenses-and-copyright.

a. You may seek permission from the original copyright holder of Figures 2 and 5 to publish the content specifically under the CC BY 4.0 license.  

Additional Editor Comments:

Editor Comment:

The paper provides an intriguing and informative analysis. The referees have highlighted several concerns that require thorough attention and addressing from the authors. In addition, to further enhance its robustness, I would suggest that the authors incorporate lagged dependent variables in the regression model. This inclusion can account for the likelihood that previous instances of hate crimes contribute to subsequent hate crimes, potentially overshadowing the effect of local deprivation. Furthermore, it would be valuable for the authors to explore the possibility that the local deprivation effect can be understood within the context of micro-dynamic models that incorporate social networks as a mechanism for the diffusion of civil unrest behavior. For instance, Braha, D. (2012) examines this topic in their work "Global civil unrest: contagion, self-organization, and prediction" published in PloS One, 7(10), e48596. Discussing and referencing this work would provide additional insight into the dynamics of local deprivation and hate crime behavior.

Reviewers' comments:

Reviewer's Responses to Questions

**Comments to the Author**

1. Is the manuscript technically sound, and do the data support the conclusions?

Reviewer #1: Partly

Reviewer #2: Partly

Reviewer #3: Partly

2. Has the statistical analysis been performed appropriately and rigorously? 

Reviewer #1: Yes

Reviewer #2: No

Reviewer #3: Yes

3. Have the authors made all data underlying the findings in their manuscript fully available?

Reviewer #1: No

Reviewer #2: No

Reviewer #3: No

4. Is the manuscript presented in an intelligible fashion and written in standard English?

Reviewer #1: Yes

Reviewer #2: Yes

Reviewer #3: Yes

5. Review Comments to the Author

Reviewer #1: This paper posits that community deprivation increases the risk of right-wing radicalization. It argues that previous literature has focused on the wrong level of analysis, by analyzing the deprivation and radicalization at the individual level. The analysis is based on disaggregated data for neighborhoods in England between 2015-2019.

The paper deals with an interesting question that merits empirical investigation. However, I believe that the theoretical arguments and empirical analysis could be made stronger. Below I list some comments, suggestions, and questions that I hope the authors may find useful.

One of the central arguments of the paper is that the local neighborhood is the main reference point for individuals and acts as a basis for community identification. Individuals who observe high deprivation in their local environment are then expected to consider their own community as neglected by the state and through their political dissatisfaction they may become more susceptible to extremist right-wing ideologies.

If individuals indeed strongly identify with their local community, it is unclear why one would expect that observing deprivation within one’s own community would lead to an increase in far-right hate crime within that same community. Especially since the communities included in the analysis are very small, comprising only between 400 – 1,200 households. Currently, this assumption is not explicitly discussed, except for one sentence on p.7 in which the authors refer to a policy document that argues hate crimes tend to be perpetrated locally. I believe the paper would benefit from making this assumption more explicit and from strengthening the theoretical arguments and references to empirical work to back it up.

The authors create three indices of local deprivation. When analyzing the results, they find that the income deprivation index outperforms education deprivation and living environment deprivation. The Appendix provides details on the indicators that are included in these indices. One of the indicators included in the income deprivation index is “Asylum seekers in England in receipt of subsistence support, accommodation support, or both”. I wonder to what extent this indicator is driving the estimated coefficients on the income deprivation index, i.e., it makes sense to expect more right-wing hate crime in local neighborhoods that host a higher share of asylum seekers. It would be interesting to see a robustness check in which this indicator is dropped from the income deprivation index (and perhaps included as a covariate).

The extent to which individuals identify with their local neighborhood likely strongly varies across individuals and neighborhoods. One element that might influence such identification could be the level of ethnic diversity. The authors currently control for the Black and Minority Ethnic Population share. Would there be a way to leverage the heterogeneity in this variable (or other variables the authors may think of), e.g., looking at an interaction with the deprivation indices?

Figure 2a shows that a substantial part of the local neighborhoods in the sample report zero hate crimes for the entire study period. In the appendix the authors say they think it is likely that some areas fail to report hate crime statistics to the Home Office. Would it be possible to provide more detailed information on the number of neighborhoods that reported zero hate crimes, and provide descriptive statistics of the covariates and deprivation indices for these neighborhoods compared to the others that do report hate crimes?

If the authors suspect that hate crime statistics are incomplete, would it be possible to conduct a robustness check relying on ACLED data – which is collected in a similar manner for the whole country and possible to link to local neighborhoods? In addition, this would allow to look at different types of conflict. It is not entirely clear why you would expect exposure to deprivation to lead to right-wing hate crimes in particular? One might expect a positive correlation between neighborhood deprivation and crime/conflict in general. ACLED data would allow to look at this. And, since ACLED records the actors that are involved in conflict events and provides some qualitative information on what happened, it might be possible to identify right wing hate crime and look at its relative importance as compared to overall crime in a neighborhood.

Data underlying the deprivation indices were collected between 2012-2013. The analysis looks at hate crimes committed between 2015-2019. I am wondering if it would be possible to say something about the timing between witnessing deprivation and committing hate crimes? E.g., is there any theoretical work suggesting what time frame matters, how long one should be exposed to deprivation before moving to committing hate crimes? Is there anything that can be done empirically? E.g., by collecting info on deprivation from earlier periods as well & looking at longer historical deprivation?

Could you clarify for what time period the covariates were collected?

The analysis includes fixed effects at the level of police force districts. It would be interesting to see more information on the police force districts. How large are they, how many of them are there, and how do they relate to the local neighborhoods / LSOAs? Perhaps you could show this on a map?

Could you add explanatory notes to the tables? I wonder for instance if the standard errors are clustered at the level of the LSOA, or at the level of the police force district?

It would be interesting to see full results reported (maybe in appendix), i.e., including estimated coefficients on the covariates.

In terms of predicting future hate crimes there is a relatively small difference between a model that only includes demographic and structural information compared to a model that additionally includes information on deprivation (a move from explaining 9% of variance to 13%). Also visually, in Figure 5, it is hard to see a difference between the full model (b) and the null model (c). Then, in terms of policy implications, to what extent could one expect that efforts to reduce local deprivation may reduce hate crime?

While replication data is not yet made available, the authors do note that it will be made available on Harvard Dataverse upon publication.

The resolution of the Figures in the current PDF version could be improved.

Reviewer #2: Dear author(s),

I enjoyed reading your manuscript. Ultimately, however, I felt that some of your arguments need more development, and I could not understand with clarity some of your measures and methodological choices. Consequently, I was not sure what to learn from your findings, beyond the fact that neighborhoods are important and that deprivation is statistically associated with deprivation.

In this review I document my concerns, as well as suggestions on how to address some of them. Please take my suggestions just as recommendations.

GENERAL COMMENTS:

1. (p. 3, para 1) Personally, I would avoid framing individual-level research as the “wrong unit of analysis.” At least I would defend that statement very carefully. In any case, I find that these kinds of dichotomies can often be a fallacy. Individual and community-level research are arguably both important, perhaps for different reasons, but that importance can co-exist.

a. In the case of your research study, some individual-characteristics are important correlates and/or causes of right-wing extremism (e.g., certain political attitudes), while certain places may also draw that type of violence from its characteristics.

b. You could argue convincingly that one level is more important, but that any level is “wrong” is a strong statement that would require more support.

2. Your front-end was well-written and clear, but I also found it underdeveloped. For example, consider the UK Prevent scheme. To be clear, I understand that the authors’ have a different perspective of the scheme, or that my interpretation could be misguided, though I am sharing my perspective for reflection. It begins with the name, Prevent, suggesting that the framework could help curtain terrorism. Second, it suggests that being a “vulnerable population” is a condition for being a “sympathizer,” which is a condition for being a “terrorist.” The very terminology portraits terrorists as naïve individuals who are passively enticed to a narrative of violence because of their vulnerability. That might be the case for many terrorists, but is it for all of them? Aren’t there any active terrorists? Those who are not vulnerable at all, or who are cynical about the very ideologies they purportedly endorse.

Most importantly, the scheme implies that a productive path for prevent terrorism is to scrutinize vulnerable groups and, most specifically, individuals who sympathize with a certain ideology. I understand that vulnerability could be statistically associated with terrorism, but is it a condition for it? How often violent extremists do not fit under our definition of “vulnerable?” How often are sympathizers not terrorists? What would be the outcome of leveraging a group’s vulnerability and political attitudes to identify terrorism? Wouldn’t we simply run the risk of add burden to these vulnerabilities and attitudes?

This scheme looks to me as an under simplification of a complex issue. It implies conditionality out of possibly theoretical associations. I would be very careful here, particularly because this kind of rationale shares the outline of an ecological fallacy. Ultimately you did not convince of the value of using neighbors for predicting violent extremists. It is true that you show an association, but that association is nowhere near as strong or deterministic to warrant a targeted intervention. In fact, the types of targeted interventions that your framework seems to endorse could very well increase the deprivation and marginalization of the communities you investigate.

To be clear, I am not arguing that these issues are major flaws of your study. Of course it can be productive to explore the links between neighborhoods and extremists, but how? Under which circumstances? You tough on those subjects, but you offer no depth or response.

a. You had a range of other statements which, I felt, were overstated. I listed some in my specific comments.

3. I had some methodological concerns, most of which were minor.

a. (Minor concern) You use LSOAs as a proxy for neighborhoods (or communities). Is that reasonable? I am fine with that decision, though I feel you should defend it a little.

b. (Minor concern) There are time mismatches between your data sources, which could introduce issues given the time-sensitivity of the issues you explore. Data on deprivation is from 2012-2013, on hate crimes is from 2015-2021. That could introduce bias, and the authors should make it clear if they find this bias is negligent.

c. (Major concern) Throughout your front end you speak about local deprivation, which read to me as poverty or marginalization. However, you describe your measure as one about relative deprivation. Relative and absolute deprivation are very different measures and concepts, with very different implications for your findings and results. In fact, from reading your Supporting Information you seemed to include some measures of relative deprivation (i.e., inequality), and some of absolute deprivation (i.e., poverty). That distinction should be clear, and should be in the text itself, as opposed to a SI. It is one thing for a neighborhood to be very poor and isolated, and it is another thing for a neighborhood to be very unequal, meaning it has both rich and poor individuals (relative to one-another). I suggest you clarify what you mean by deprivation.

d. (Minor) The fond size of your figure 2 is too small. The figure was hard to read, and hard to compare across indicators.

e. (Minor) Unclear to me what was the benefit of splitting your sample and to use estimates from earlier years to predict future years. I recommend you clarify the benefit, or that you avoid that part of your analysis. Why not simply use all cases to develop your model?

f. (Minor) In Figure 3 it was unclear why you separated Income Deprivation from Loving environment and education deprivation. I assume it was to avoid clutter in the figure. If that was the case, I would point out that, in my view, have population data instead of data about a representative sample from which you are trying the extract inferences. In other words, your coefficients are not sample estimates of an unknown population parameter, but instead reflect the population parameters themselves. Though your parameters are still subject to several kinds of errors, they are not subject to sampling error, which is the only kind of error that is informed by the p-value. Therefore, your p-values and significances carry relatively little importance. For this reason, consider removing the confidence intervals from Figure 3.

SPECIFIC COMMENTS:

4. I appreciated the simplicity and objectivity of your research question, which makes it appealing to the broader scientific audient of PLOS ONE.

a. In fact, all aspects of your study are clear and parsimonious, including your analysis and text. Overall a very enjoyable article to read.

5. (p. 3) About the passage: “We provide a first study to systematically examine this relationship at the level of neighborhoods,” I suggest you rephrase. I am not entirely sure what you mean by “systematic” here, but it is very possible that your study is not the first—which does not take from your merit.

6. (p. 4) About: “We believe that a similar process also applies,” you include citations, which suggests the argument is not originally yours. I suggest you either clarify or rephrase to “A similar process could also apply.”

7. (p. 4) About: “We posit that the local neighborhood is the main reference point for individuals’ comparisons, shaping their perceptions and beliefs about outcomes and distributions.” Why not their street? Or their family? I understand that neighborhoods are important, but are you confident that they are the “most” important? That is a strong statement. Consider toning it down, (e.g., “is a key reference) or supporting it more in terms of it being the “most important.”

8. (p. 13) Which “theoretical mechanism” are you referring to specifically? Unclear. Your discussion and conclusion is generally underdeveloped.

Reviewer #3: I enjoyed reading this paper on neighbourhood deprivation and hate crime. I am convinced by the central argument and I think it should be published. I have just some comments that might change the analysis.

First though, I note that the authors have NOT made data and code available at time of submission. This means I have to answer NO to the data availability question above. Also, it makes for less good peer review, since I can't scrutinise the code or run the analyses in writing my report. I would strongly urge the authors to always prepare the data and code archive to go along with the manuscript for evaluation. In my discipline, and in many journals, this would actually be a condition of submission. If the authors are worried about precedence, they should publish a preprint version at the same time.

Now, for my main comment, which is that I don't understand the status of the three separate deprivation measures. The authors do not report the intercorrelations between these. If the intercorrelations are very high, and the authors consider them multiple measures of the same thing, then they should just average them. (Or use the overall IMD score, which effectively does this). If on the other hand they consider them to be potentially important independent components, then they should also evaluate models containing more than one of them at a time (e.g. in table 1). The way I would do this is to run AIC-based model selection (such as R package MuMIn) on the set of the covariate-only model, the null model, then all models including the covariates plus every combination of the three deprivation measures. It may be that the one with just the income deprivation measure wins, but this would be interesting and also worthy of substantive discussion (i.e. income is the prime factor in the kind of deprivation that matters for hate crime).

More generally, reporting AIC and AIC change is a much better approach than comparing R2 to show that variables improve model fit.

Specific comments

p. 5. "The individuals who carry out attacks are necessarily relatively more deprived, even if their local community is deprived." - is this sentence missing a 'not'?

p. 6. "Hate crimes differ from other criminal offences in that they express a number of socio-political objectives by...." Do the authors have any concerns about classification bias by police? i.e. that the same offence committed in a deprived neighbourhood is more likely to be classified as a hate crime than when committed in an affluent neighbourhood? (there could be a host of reasons why this is true, including the ethnic composition of the people there, etc.). Is it worth discussing this possibility, or at least saying something about how hate crime status is determined. It must be a judgement call. The authors already consider the possibility that different police forces might classify differently, which suggests a degree of classification latitude that could cause endogeneity problems.

p. 7. "We use the indices for deprivation in income, education, and environment as complimentary measures of local deprivation" - How well correlated are these measures? Are they so well correlated that they should just be considered as the same information, or are you trying to argue that they capture something distinct? Their inter-correlations should be reported.

p. 9. Table 1. I would be much happier with the analysis if the authors reported AIC instead of R2. The critical question is how much AIC goes down by inclusion of deprivation indices.

Also, as above, I fail to see the logic of models in which one at a time of the three deprivation measures (but not the others) is included. The authors should establish the intercorrelations between the three deprivation measures a priori. If this is very high, their argument would be better served by making one measure out of the three. (I believe in fact that there is an overall IMD measure already provided). If it is not very high and they make a unique contribution, then they need to try models including more than one of the three. You could also using AIC-based model selection to create the optimally predictive model (e.g. using the MuMIn package in R). This would give a sense of which is most important, income, education or environment.

Figure 6. I like this figure a lot, but I wonder if it would be more legible with two side-by-side panels, one for the without deprivation model, and one for the with deprivation. And, it would be helpful to have the line of best fit as well as the y=x line in each case; it should be appreciably flatter without deprivation. Are all three deprivation measures used for the with-deprivation predictions here?

p. 11. I heartily agree with the authors' suggestion that 'upstream' relief of deprivation is likely to be effective at reducing hate crime (among other things). But, given the scatter in figure 6, it is a very blunt instrument. That is, if your objective were ONLY to reduce hate crime, levelling up would probably not be a cost-effective way of doing it, since the relationship between deprivation and hate crime is messy. The way I tend to think about these things is that we should be reducing deprivation for many reasons; reducing hate crime is, plausibly, one of the many benefits, but not by itself a justification for doing so.

6. PLOS authors have the option to publish the peer review history of their article (what does this mean?). If published, this will include your full peer review and any attached files.

Reviewer #1: **Yes: **Nik Stoop

Reviewer #2: No

Reviewer #3: No

---

## [Author Response · Author response to Decision Letter 0]

14 Jul 2023

[Note: We have also uploaded a revision memo as a separate file with the manuscript. We reproduce the text below, but would encourage you to look at the file since some formating does not transfer correctly as plain text.]

Revision Memo: PONE-D-23-08904 "Local deprivation predicts right-wing hate crime in England"

We refer to your email of 12 June regarding our submission PONE-D-23-08904 "Local deprivation predicts right-wing hate crime in England". We are very grateful for the invitation to resubmit a revised version, and we appreciate the efforts of the editor and reviewers for engaging so thoroughly with the manuscript. 

We respond below to the main concerns raised, ordered by topic, with our responses and comments in italics. The reviewers also made a number of smaller comments and suggestions that we have tried to incorporate to the best of our ability in the revisions. We believe the revised version has been notably strengthened as a result of the revisions made in response to the comments. 

Please do not hesitate to contact us if we can help with any additional information. 

Sincerely yours,

The authors 

A. Theory

A1) Clarification of relationship between community deprivation and attacks

R1: “If individuals indeed strongly identify with their local community, it is unclear why one would expect that observing deprivation within one’s own community would lead to an increase in far-right hate crime within that same community. …. Currently, this assumption is not explicitly discussed …. I believe the paper would benefit from making this assumption more explicit and from strengthening the theoretical arguments and references to empirical work to back it up”. On a related issue, R2 warns against “framing individual-level research as the ‘wrong unit of analysis’ … and [how] these kinds of dichotomies can often be a fallacy 

We have expanded our discussion of why we would expect to find a relationship between local deprivation and attacks. In brief, we argue that hate crimes typically target specific communities or individuals based on their characteristics. There is empirical evidence demonstrating that perpetrators of hate crimes tend to carry out violence within their local neighborhood. There are a number of reasons why perpetrators often seek out victims within their immediate vicinity, notably 1) it is easier to identify targets they hold prejudice against in the local community, 2) access, planning and execution is typically easier for local targets, 3) local targets often have symbolic significance for both perpetrators and targeted communities, and 4) local hate crimes will often receive more attention in the immediate community, which in turn can be helpful for generating fear, exerting control, and recruiting to the cause. We agree with R2 that it is unhelpful to draw a false dichotomy between individual level analysis and other levels of aggregation. We have revised the text to emphasize more clearly the added and complementary value of local level analysis.

A2) Relationship to relative and absolute deprivation. 

R2 asked us to clarify the relationship between relative and absolute deprivation in the theoretical framework. We appreciate that our previous discussion may have been confusing on this point, and we have clarified our conceptualization and actual measures of deprivation in the manuscript. We emphasize that what we think drives the key relevant comparisons here are differences between neighbourhoods, not within community differences. 

A3) Discussion of UK prevent scheme

R2 notes that … “This scheme looks to me as an under simplification of a complex issue … the scheme implies that a productive path for prevent terrorism is to scrutinize vulnerable groups and, most specifically, individuals who sympathize with a certain ideology. … What would be the outcome of leveraging a group’s vulnerability and political attitudes to identify terrorism? Wouldn’t we simply run the risk of add burden to these vulnerabilities and attitudes?”

We appreciate the comments on the UK Prevent scheme. We also share many of the concerns expressed over the scheme, and we try to be more upfront about this in the manuscript. At the same time, this scheme has clearly guided policy and discussion, and as such it is useful to discuss our approach in comparison to this. We did not intend to apply the term “vulnerable” as a value judgement or to underplay individual agency, rather we simply seek to examine if features discussed in this context can help identify local contexts with a higher probability of radicalization and observed attacks. We primarily discuss the potential influence of deprivation in our manuscript rather than measures against potentially radicalized individuals or communities. We have revised the discussion so as to be much clearer on this and avoid potential misunderstanding. 

A4) Diffusion and network effects

The editor noted that “it would be valuable … to explore the possibility that the local deprivation effect can be understood within the context of micro-dynamic models that incorporate social networks as a mechanism for the diffusion of civil unrest behavior … [discussing and referencing] Braha (2012) … would provide additional insight into the dynamics of local deprivation and hate crime behavior.

We entirely agree that networks and diffusion are important components in accounting for observed unrest and hate crime. We have added a brief discussion and referenced work on the diffusion of political violence. However, a full examination of this would require a much more complex analyses of the role of “local fundamentals” vs “position”, and ideally consider detailed analyses of individual connections using actual data rather than proxies based on geographical distance. We also note that very strong diffusion or emulation effects could plausibly attenuate the direct local relationships, and this appears to be the case in many studies of organized political violence. 

A5) Time ordering in exposure and response

R1 and R2 noted that gap in time between deprivation indices for 2012-2013 and hate crimes committed 2015-2019. “… would [it] be possible to say something about the timing between witnessing deprivation and committing hate crimes? E.g., is there any theoretical work suggesting what time frame matters, how long one should be exposed to deprivation before moving to committing hate crimes?” (R1)

We use prior deprivation data since we want to ensure that the deprivation data is clearly prior and is not likely to arise from as a result of or influenced by our outcome. There is little direct guidance in previous work on the likely timing of exposure and potential impact on attacks, but we believe the lag length in our case is a reasonable length to reflect the impact of past exposure to deprivation. In practice, there exists considerable persistence in deprivation over time, and this is supported by the data on deprivation published by Ministry of Housing (see for example Indices of Deprivation: 2019-2015 at https://dclgapps.communities.gov.uk/imd/iod_index.html). We have revised our discussion so as to make clearer the assumptions guiding our analysis, the limits related to actually considering sensitivity to timing in the current data and our analysis, and the value of future more detailed research on this. 

A6) Conditional effects

R1 notes that “The extent to which individuals identify with their local neighborhood [might vary by] … ethnic diversity. … Would there be a way to leverage the heterogeneity in this variable (or other variables the authors may think of), e.g., looking at an interaction with the deprivation indices?”

We think it is plausible that such heterogeneity may be present in the observed data, and we have conducted analyses where we interact ethnic diversity, proxied as the local population share with BAME background, with our indicators of deprivation. We find that ethnic diversity does not substantially moderate the effects of living environment deprivation and education deprivation, but that high levels of diversity may mitigate the effect of income deprivation. However, we ultimately feel that extending the analysis further along these lines is somewhat removed from our main focus in the manuscript, and we thus report in an appendix without an extended discussion in the main manuscript text. We think a fuller investigation of this would amount to a different project and better pursued in a separate manuscript. We also note that measuring conditional interactions or moderators reliably demands a great deal more of the data than main effects, and could benefit from more extended data than we have at hand now. 

B. Data and empirical analysis

B1) Lagged dependent variables and autoregressive trends

The editor noted that “to further enhance its robustness, I would suggest that the authors incorporate lagged dependent variables in the regression model …[to] account for the likelihood that previous instances of hate crimes contribute to subsequent hate crimes, potentially overshadowing the effect of local deprivation”. 

We appreciate the point made, and we entirely agree that hate crime data are likely to show considerable persistence and autoregressive trends. Including a lagged dependent variable would be natural if we had panel data, but unfortunately, we do not – the statistics for hate crimes that we rely on, and the time gaps between releases of the Indicators of Deprivation, do not allow for a time-series setup. Therefore, we are essentially only able to conduct a cross-sectional analysis, and it is not possible to include a lagged dependent variable in the regression model. We now explicitly clarify this in the revised version of the manuscript. We also do not propose to measure feature/variable importance by contribution to fit in-sample, which is prone to overfitting, but rather contribution to out-of-sample prediction (see B3 below). 

B2) Measures and specific components

R1 notes that the income deprivation index [includes] ‘Asylum seekers in England in receipt of subsistence support, accommodation support, or both’. I wonder to what extent this indicator is driving the estimated coefficients on the income deprivation index … It would be interesting to see a robustness check in which this indicator is dropped from the income deprivation index (and perhaps included as a covariate).” 

We have looked at the available data and the Home Office statistics to verify if we could find any plausible proxies. There is unfortunately no publicly available sub-national data, so we are unable to probe how sensitive the findings may be to this. We believe it is likely that the IoD obtained these data directly from the Home Office at LSOA level. The only way which we might be able to obtain these data would be to submit a FOI request, which may or may not be successful. This is not feasible within the deadline for the resubmission, and we feel that this goes beyond the main priorities for the resubmission. The fact that we consider alternative indicators for deprivation other than income, which do not include asylum seekers in receipt of support, makes it unlikely that this component alone would dominate the results. However, we now acknowledge this limitation in the main text and explicitly flag this as an area for future study. 

On a similar note, R3 argues that the status of the deprivation measures must be clarified, and that we should “consider … average[ing] them … or evaluate models containing more than one of them at a time”

We now report the correlations in the Supplementary Appendix. We understand the case for trying to aggregate the information into a combined deprivation index, but to do this systematically should ideally be based on the raw inputs of the indices, rather than averaging across aggregate indices with different scales/metrics. Ultimately the regression models are all in-sample, and they are intended to evaluate the conditional associations/causal effect approximations in a simple manner. For this particular purpose, including all deprivation measures in one model would make the interpretation more difficult. For prediction, on the other hand, we actually use a random forest model that includes all the deprivation scores in the same models for computing the predictions of future levels of hate crime. We have now clarified this issue in the discussion in the manuscript.

B3) Role of prediction in model evaluation

Rs 1 and 2 raised some question about the value of splitting the data, emphasis on prediction, and added value of the model with deprivation. We have now revised the text so as to clarify why cross validation is an important check on overfitting to observed data. It also helps emulate the potential to use measures for out of sample forecasts. We have also clarified that even if the variance accounted for may seem small in absolute terms, there is a near double increase for the deprivation model. We think this demonstrates the added value, although we try to be explicit on not overstating the absolute predictive power. 

B4) Clarify data

Rs1, 2, and 3 asked for clarification for many aspects of the data, including the timing of the covariates, and the size/number of police force districts used for fixed effects and relationship to local LSOAs, and the district reporting no hate crimes. We have revised our discussion so as to provide more details explicitly on the issues raised. We also discuss in more detail the potential bias from underreporting of hate crimes (which under plausible condition would likely attenuate the reported relationship with deprivation). The Home Office informed us about the police districts that did not submit any reports in specific years (where the “zero hate crimes” is the result of a reporting issue), and these districts were removed from all analyses to begin with. In addition, we report results with all units with zero hate crimes removed from the analyses in the Supplementary Appendix, referenced in the main manuscript.

B4) Clarify the analyses reported

Rs 1,2, and 3 called for more details in explanatory notes (e.g., types of standard errors), and we have implemented these in the revised manuscript. We also report AICs in addition to R2, as requested. 

B5) Figures/image resolution

R3 suggested to revise Figure 6 to include two side-by-side panels, one for the without deprivation model, and one for the with deprivation, with a best fitting regression line. We have also improved the resolution of the images in the manuscript. The small font sizes are the result of downscaling in MS Word and potentially only PDF conversion. This will not be an issue in the published version of the manuscript using the separate images. 

B6) Supplementary analyses

R1 suggested a robustness check relying on the ACLED data. We appreciate this suggestion, but we are unable to use ACLED, since the ACLED data for the United Kingdom are limited to the period between 2020 and 2022, cutting significantly our sample. (Moreover, ACLED has a restrictive terms of use license, which does not allow for comparisons to other data sources, see https://acleddata.com/acleddatanew/wp-content/uploads/2022/06/ACLED-Terms-of-Use-Attribution-Policy_V2_8June2022.pdf). In general, with news-based event data (like ACLED and GTD) we can expect the coverage of the events to be more limited and arguably thus inferior to our administrative data. We believe the selection problems from media reports are likely to be worse than the potential problems arising from differences in reporting by administrative agencies. We discuss in more detail how to best attempt to mitigate issues arising from underreporting.

C. Broader implications/policy

Some of the reviewers raised issues about the potential broader implications and actionable policies that our manuscript could be seen as supporting. We have tried to be careful to be modest in our discussion of policy implications, in particular since we have not studied intervention to reduce deprivation or inequality. However, we note the reductions in child poverty under New Labour as a possible example of how reforms seeking to reduce inequality could have wider impacts or dividends on violence, even if reducing violent attacks by itself is the main target of policy proposals. 

D. Journal Requirements:

D1) Compliance with style requirements, including file names. 

We have carefully reviewed the guidelines to ensure that the manuscript and file names follow the PLOS One guidelines. 

D2) Role of funders

Please state what role the funders took in the study… Please include this amended Role of Funder statement in your cover letter; we will change the online submission form on your behalf.

We have included the suggested text in our cover letter: "The funders had no role in study design, data collection and analysis, decision to publish, or preparation of the manuscript." 

D3. Data Availability statement 

We have made our data available on the Harvard Dataverse at https://doi.org/10.7910/DVN/GSRPQY.

D4) Re license/copyright for maps used for Figures 2 and 5 

These images have been generated by shape files that are publicly available from the UK Office for National Statistics. They are supplied under the Open Government Licence. We have added the source and license as “Office for National Statistics licensed under the Open Government Licence v.3.0”.

D5). … include captions for your Supporting Information files at the end of your manuscript, and update any in-text citations to match accordingly. 

We have now implemented this.

---

## [Editor Report · Decision Letter 1]

19 Jul 2023

Local deprivation predicts right-wing hate crime in England

PONE-D-23-08904R1

Dear Dr. Gleditsch,

We’re pleased to inform you that your manuscript has been judged scientifically suitable for publication and will be formally accepted for publication once it meets all outstanding technical requirements.

Best,

Dan Braha

Academic Editor

PLOS ONE
---

## [Editor Report · Acceptance letter]

24 Jul 2023

PONE-D-23-08904R1 

Local deprivation predicts right-wing hate crime in England 

Dear Dr. Gleditsch:

I'm pleased to inform you that your manuscript has been deemed suitable for publication in PLOS ONE. Congratulations! Your manuscript is now with our production department. 

Kind regards, 

on behalf of

Professor Dan Braha 

Academic Editor

PLOS ONE